# Not All Workers Experience Equal Sleep Changes: Insights from the “WorkInCovid” Project

**DOI:** 10.3390/clockssleep7010013

**Published:** 2025-03-10

**Authors:** Sergio Garbarino, Antonella Bodini, Saverio Sabina, Carlo Giacomo Leo, Pierpaolo Mincarone, Antonella Rissotto, Stanislao Fusco, Roberto Guarino, Maria Rosaria Tumolo, Giovanni Luigi Tripepi, Egeria Scoditti, Nicola Magnavita

**Affiliations:** 1Department of Neurosciences, Rehabilitation, Ophthalmology, Genetics and Maternal/Child Sciences (DINOGMI), University of Genoa, 16132 Genoa, Italy; 2Post-Graduate School of Occupational Health, Department of Life Sciences and Public Health, Università Cattolica del Sacro Cuore, 00168 Rome, Italy; 3Institute for Applied Mathematics and Information Technologies “E. Magenes” (IMATI), National Research Council (CNR), 20133 Milano, Italy; antonella.bodini@cnr.it; 4Institute of Clinical Physiology (IFC), National Research Council (CNR), 73100 Lecce, Italy; saverio.sabina@cnr.it (S.S.); carlogiacomo.leo@cnr.it (C.G.L.); roberto.guarino@cnr.it (R.G.); egeria.scoditti@cnr.it (E.S.); 5Institute for Research on Population and Social Policies (IRPPS), National Research Council (CNR), Research Unit of Brindisi, 72100 Brindisi, Italy; pierpaolo.mincarone@irpps.cnr.it; 6Training and Welfare Unit, National Research Council (CNR), 00185 Rome, Italy; antonella.rissotto@cnr.it (A.R.); stanislao.fusco@cnr.it (S.F.); 7Department of Biological and Environmental Sciences and Technology (DISTEBA), University of Salento, 73100 Lecce, Italy; mariarosaria.tumolo@unisalento.it; 8Institute of Clinical Physiology (IFC), National Research Council (CNR), 89124 Reggio Calabria, Italy; giovanniluigi.tripepi@cnr.it

**Keywords:** sleep health, excessive daytime sleepiness, telecommuting, COVID-19 pandemic, sleep quality, chronotype, weight gain, depression

## Abstract

The COVID-19 outbreak has changed work organization in favor of a working from home (WH) modality. We examined the association of WH during the pandemic with sleep health in workers of a public research organization. An online cross-sectional survey in 2022 at the National Research Council of Italy collected information on sociodemographics, work characteristics, and sleep pattern during WH compared with before WH. In the whole sample (*n* = 748), total sleep quality did not significantly change. Excessive daytime sleepiness (EDS) decreased during WH. Total sleep quality increased during WH in poor sleepers, while it decreased in good sleepers. The morning chronotype was protective against sleep worsening in poor sleepers. Risk factors were depression in poor and good sleepers, and increased daytime sleepiness and body weight gain in good sleepers. These findings emphasize the importance of baseline sleep pattern in shaping the impact of WH on sleep.

## 1. Introduction

The epidemic caused by the SARS-CoV-2 virus, known as COVID-19, was the first pandemic to occur in a globalized and connected world. For this reason, it represented an extraordinary opportunity to study workers’ response to the abrupt changes in work organization that were imposed by the pandemic. In the run-up to the pandemic, research rightly focused on safety procedures and case treatments, especially in frontline healthcare workers whose efficiency was essential to deal with the pandemic [1]. Today, we can carefully re-evaluate the effects of the sudden change in the methods and times of production.

The need to control the spread of infection led all countries in the world to decree measures to limit the mobility of citizens. For most businesses, teleworking or working from home (WH), a work arrangement based on work flexibility and autonomy, was the only way to continue production during the lockdown.

WH was actually a widely spread experience for at least fifty years, when, especially in Anglo-Saxon and Northern European countries, the idea was to limit think-tank by entrusting employees with tasks to be carried out at home and keeping in touch through telephone. The information revolution in the United States of America gave a strong boost to teleworking. The first to work remotely were programmers, who lived in Silicon Valley in California but operated throughout the country. Subsequently, many companies have found it convenient to use remote work for part of their working hours [2]. This method offers advantages for companies, which save space, but also for workers, who can better organize extra-work activities and save travel costs and time [3,4]. However, teleworking has the limit of reducing contacts between workers thus, in the long term, hindering debate and growth. For this reason, before the pandemic, many companies resorted to flexible working, maintaining close interpersonal contacts for part of the working hours. Before the pandemic, the organization of teleworking presented critical aspects especially related to the leadership style and the work required outside working hours [5].

The explosion of the COVID-19 pandemic forced many companies to start teleworking without any preparation. This unexpected experience was seen by some in a very positive way, as an opportunity to have more time to dedicate to the family, improve relationships with family members, and ensure a safe home environment. In other cases, numerous stress factors added to the fear of infection, economic problems, and uncertainty about the future that are typical of a pandemic. These factors included inexperience in this way of working, the lack of a work environment with ergonomically acceptable requirements and sometimes with a single home computer to share with other family members, the lack of contact with colleagues and information on the work to be carried out, and the extension of work requests even beyond traditional working hours. It was thus possible that the advantages of WH were balanced and sometimes overcome by significant stressors [6]. These situations can be considered stressful, and the European Parliament has drawn the attention of the legislator to this topic, to adapt workplace safety legislation to unforeseen conditions [7].

A possible sensitive indicator of workers’ wellbeing or discomfort during the lockdown teleworking experience is sleep [8]. Sleep is an active neurobehavioral state vital for physical and mental health and social wellbeing, and is therefore recognized as essential to health, like nutrition and physical activity [9,10]. Multiple sleep characteristics (or dimensions) are indicators of sleep health, including sleep duration, continuity or efficiency, timing, alertness/sleepiness, and satisfaction/quality, which can be evaluated in relation to health outcomes both independently and in combination [11]. Overall sleep and its dimensions are driven by biological factors (e.g., genetics, health status) as well as environmental and societal factors [11,12].

Over the last few decades, inadequate sleep due to insufficient sleep duration (less than the minimum requirement of 7–8 h per night for adults) and/or quality has become an increasingly prevalent problem in industrialized economies [13]. Poor sleep can be the result of a complex network of behavioral, psychological, environmental, and societal influences. In working adults, workplace-related and organizational factors such as work schedule (e.g., shift work), irregular working hours, overtime work, high work demands, interpersonal conflict, stressful targets and time pressures, commuting, job insecurity, as well as work–life balance have been found to be associated with an increased prevalence of sleep problems [13]. Other potential contributing factors encompass mood disorders [14] as well as unhealthy lifestyle (e.g., smoking, physical inactivity, poor diet) [15,16,17].

Insufficient sleep, circadian misalignment, and untreated sleep disorders are determinant risk factors for major chronic diseases, such as cardiovascular and neurodegenerative diseases, obesity, diabetes, cancer, and mortality [18,19,20], as well as for cognitive functioning and emotional regulation [21,22,23,24]. Poor quality and quantity of sleep may lead to sleepiness, fatigue, reduction in memory consolidation, and cognitive impairment [23] and, as such, influence job performance and safety [25,26]. Basic mechanisms linking poor sleep and sleep disorders with disease risk include a chronically deregulated immune response, heightened inflammatory reaction, and oxidative stress, which are driving processes in the development of cardiometabolic, neurodegenerative, and mental illnesses [18,27,28]. Poor sleep has an economic burden due to its negative effects on health, thus compromising not only global health but also the national health budget of all countries [29].

Sleep is one of the health indicators that was influenced by the COVID-19 pandemic, which abruptly changed our life and work patterns worldwide. Sleep was often compromised in those affected by COVID-19 [30,31,32]. Sleep impairment is one of the most frequent symptoms in long COVID syndrome [33,34].

In the first phase of the pandemic, poor sleep quality significantly associated with anxiety was demonstrated in healthcare workers who had been infected or had unprotected contact with infectious patients [35], or in first-line workers who had to fight an unknown disease, with limited therapeutic resources [36]. Subsequently, through the repetition of the pandemic waves, health workers’ sleep problems resulted from the excessive workload and compassion fatigue [37,38], isolation, and lack of time for physical activity and meditation. Understandably, healthcare workers have been carefully studied, but much less information is available on the well-being and sleep quality of other workers. At the moment, there is a lack of data on sleep patterns in workers who were forced to telework during the pandemic [39].

This lack of studies could have a significant impact on public health, because the number of workers who have telecommuted is much higher than that of healthcare workers engaged in COVID-19 medical centers. It is very important to know what impact WH has had on sleep to be better prepared in the event of a future epidemic, but also considering that, in the aftermath of the COVID-19 pandemic, WH policy has been increasingly recognized as a sustainable alternative to in-presence working.

Two opposite hypotheses could be formulated a priori: some of the workers would have reported a worsening in the quality and quantity of sleep during the working from home, while others would have reported a period of wellbeing and improved sleep. The literature does not authorize us to favor one of the two hypotheses.

The associated impact of WH on health and wellbeing outcomes may depend on many factors related to the individual capacity and/or willingness to manage WH, to the organizational support, the home environment, and the external challenging context of the COVID-19 pandemic [39,40,41,42]. Limited evidence reported a positive effect of WH adoption during the COVID-19 outbreak on sleep patterns, observing a reduction in social jet lag and an increase in sleep duration [43,44,45], but the effects on overall sleep quality were not univocal [46,47,48].

During the first phases of the pandemic, the prevalence of urgent concerns about managing the emergency diverted attention from the topic of “sleep”. As soon as the tension eased, however, it became clear that this topic deserved attention. Many workers had experienced for the first time the forced transition to 100% teleworking for several months and this had profoundly changed their way of working and managing family activities, so that there having been an impact on sleep was highly likely. Workers’ opinions about the new way of working were polarized, some in favor, others against. Some studies indicate that during this period, there were changes in diet, physical activity, mental health, and other aspects of family life that influence such opinions [49]. We wondered if there also was a change in sleep.

To study this topic, we decided to investigate the worker population of a public research organization in Italy with a retrospective design focused on describing the situation before the pandemic and the one after. The extent of the change brought about by the pandemic and the lockdown was such that it allowed those involved to remember their conditions perfectly, despite the time that had passed. Precisely because of the time that had passed, it was foreseeable that not everyone would respond, but only those who were keen to indicate an improvement or worsening in conditions. It was also foreseeable that the positive or negative responses could balance each other out. Once this study was constructed with these hypotheses, from the set of responses received, we isolated those who had poor quality of sleep before the pandemic, and those who had good quality of sleep before the pandemic, to evaluate changes in sleep and the potential factors associated with those changes.

The aim of this work was to verify what effect pandemic-induced teleworking had on sleep. The hypotheses we derived from our observations on workers were as follows:Workers who had better quality of sleep before WH could have a worsening in quality of sleep, due to the combined effect of anxiety related to the pandemic and the sudden and unorganized change in working methods.Workers who had poor quality sleep before the lockdown could have an improvement in sleep during teleworking, due to the reorganization of time dedicated to work and family and the avoidance of commuting.

We also assumed that workers who had experienced a change in their sleep habits would be the most willing to respond, while those who had not noticed any changes in their sleep would be less interested in our survey.

## 2. Results

### 2.1. Sample Characteristics

General characteristics of the study sample are displayed in Table 1. A total of 748 individuals participated in the survey, of which 733 subjects (98%) completed all the survey sections. The majority of the participants were women (56.8%), in the 40–49 and 50–59 age range, highly educated, residing in Central and Northern Italy, not living alone and with children, and were researchers and technologists. Comparisons with the CNR source population are reported in [50].

### 2.2. Changes in Sleep Pattern in the Whole Sample

Looking at the whole sample, subjects reported no statistically significant changes in Pittsburgh Sleep Quality Index [51,52] (PSQI) total score, indicating overall sleep quality (Table 2). Referring to the in-presence working period, the global PSQI score ranged from 0 to 17; the quartiles were 4, 5, and 6; while the mean (±SD) score was 5.2 (±2.4). A slight shift toward lower values was reported during WH (range from 0 to 17; quartiles 3, 5, and 7; mean score 5.15 ± 2.57). Despite a significant *p*-value (0.04) of the paired Wilcoxon test on the differences in the total scores, the estimated location shift is almost negligible (estimated pseudo-median: 1.5 × 10^−5^), and therefore no meaningful changes could be found. The percentage of poor sleepers (PSQI > 5) did not change significantly (from 38.7% to 36.9%, *p* = 0.37). Given the well-known impact of poor sleep quality on all aspects of health, we present the results below both for the sample as a whole and as a function of the condition (poor vs. good sleep) reported before the pandemic and the transition to WH.

Reported changes in sleep duration and sleep efficiency were statistically significant (*p* < 0.005) (Table 2). Sleep duration significantly increased during WH, with an estimated positive true location shift of 30 min (95% CI: 17.5–42.5 min). The percentage of subjects with a short sleep (≤6 h) significantly decreased (from 33.8% to 27.3%), while the percentage of subjects with an adequate sleep duration (7–8 h) increased (from 65.7% to 70.4%) as well as the percentage of subjects with long sleep duration (from 0.5% to 2.3%). However, the sleep efficiency significantly decreased by about 1.2%.

The proportions of subjects reporting an improvement or a worsening in the PSQI components during WH are presented in Table 3. As also shown in Appendix A, the percentages of participants with better conditions in terms of subjective sleep quality and daytime function increased, while the percentages of participants reporting better conditions in terms of sleep latency, sleep disturbances, and use of sleep medications decreased.

Stratified analysis showed that during WH, women had a higher total PSQI score (indicative of lower sleep quality) than men (*p* < 0.001), but the reported changes moving on to WH were not significantly associated with gender (Appendix A). Researchers reported a lower total PSQI score than technical and administrative staff (*p* < 0.001) both before and during WH, but the reported PSQI variation was not associated with the job profile (Appendix A).

Regarding excessive daytime sleepiness (EDS), the Epworth Sleepiness Scale (ESS) score range reduced from 0–23 to 0–15 but the total score did not show any significant location shift: the quartiles were 2, 4, and 6 both before and during WH, while the mean (±SD) score only decreased slightly, from 4.46 (±3.2) to 4.30 (±2.85) (Table 2). Despite this, the prevalence of reported EDS, i.e., ESS > 10, significantly decreased from 5.4% to 3.1% (*p* = 0.007).

Regarding napping habits, only 32 respondents (4%) used to take an afternoon nap before the transition to WH, with a duration up to 30 min for half of them. Twenty-one of these subjects maintained the habit with the transition to WH. Interestingly, there was a significant increase in the nap habit for 76 (13.2%) participants during the WH (*p* < 0.001).

### 2.3. Changes in Poor and Good Sleepers

As expected, PSQI values during WH were significantly associated with the baseline pre-WH levels of sleep quality (Kendall’s τ coefficient value −0.27, *p* < 0.001). However, changes in sleep quality could be observed in subgroups that defined themselves as poor or good sleepers before the pandemic. Specifically, participants with poor sleep quality before WH (n = 285) reported an improvement in sleep quality when working from home, with an estimated true location shift of −1 (*p* < 0.001) (Figure 1A), while the good sleep subgroup (n = 452) worsened their sleep quality, with an estimated true location shift of +0.5 (*p* = 0.001) (Figure 1B). Moreover, 161 out of the 285 subjects (56.5%) with total PSQI score >5 before WH improved sleep quality during WH, compared to 136 out of 452 (30.1%) of subjects with total PSQI score ≤ 5 (*p* < 0.001).

Exploring the PSQI components in the two subgroups (Appendix A), among the poor sleepers, the percentages of subjects with better conditions in terms of subjective sleep quality, sleep duration, and day-time dysfunction significantly increased during WH compared with pre-WH; contrarily, among the good sleepers, the percentages of subjects with better conditions in terms of subjective sleep quality, sleep latency, sleep efficiency, sleep disturbances, and use of sleep medications significantly decreased during WH compared with pre-WH. As relevant participants’ categories to be compared between the two groups, for each component, we considered the proportion of subjects who improved among the poor sleepers and the proportion of subjects who worsened among the good sleepers. As reported in Table 4, the proportions of participants who improved among poor sleepers were significantly higher than those of participants who worsened among good sleepers as regards sleep quality, sleep duration, and daytime dysfunction components. No between-group differences were found for the remaining PSQI components (all *p*-values were not significant).

### 2.4. Predictors of Sleep Disturbances During WH in Poor and Good Sleepers

To disentangle the effect of sociodemographic, lifestyle, and health-related parameters’ changes on the risk of sleep quality worsening (i.e., an increase in PSQI total score) in poor sleepers, the univariable logistic regression was estimated considering sleep quality as dependent variable. Statistically significant risk factors of worsening in sleep quality in subjects who had poor sleep before WH were worsened depression severity, which was indexed by higher Patient Health Questionnaire (PHQ) score [53] (OR: 3.61, 95% CI: 2.04–6.49); increased body weight (OR: 2.47, 95% CI: 1.32–4.80, reference: unchanged weight); and living in Central Italy during the WH period (OR: 2.03, 95% CI: 1.03–4.11, reference: living in Northern Italy).

On the contrary, being a morning type was a significant protective factor for sleep impairment (OR: 0.37, 95% CI: 0.14–0.99, reference: evening type). Other non-significant factors were sharp increase in sedentary lifestyle (OR: 2.71, 95% CI: 0.89–8.94, reference: decrease in sedentary lifestyle), being a technician (OR: 2.11, 95% CI: 0.89–5.10, reference: administrative staff), more than 120 working days in presence during the WH period (OR: 2.09, 95% CI: 0.93–4.58, reference: no more than 20 working days in presence). Similarly, no significant associations were observed for commuting time from home to work of more than one hour (OR: 0.40, 95% CI: 0.14–1.11, reference: no more than 15 min) and living in a house of at least 150 square meters during the WH period (OR: 0.41, 95% CI: 0.13–1.05, reference: house of less than 100 square meters).

The multivariable logistic regression model showed that an increase in PHQ score and the circadian type were the only independent predictors of increased total PSQI score (Table 5).

In good sleepers, the univariable logistic regression showed that statistically significant predictors of a worsened sleep quality (i.e., an increase in total PSQI score) in 154 out 448 (34.4%) subjects were worsened depression severity (OR: 3.6, 95% CI: 2.39–5.46), increased body weight (OR: 2.9, 95% CI: 1.87–4.45, reference: unchanged weight), increased appetite (OR: 2.47; 95% CI: 1.61–3.69; reference: unchanged appetite), decreased weight (OR: 2.05, 95% CI: 1.10–3.80, reference: unchanged weight), and worsened daytime sleepiness (OR: 2.02, 95% CI: 1.33–3.07). Having a dedicated room available for work was a protective factor (OR: 0.64, 95% CI: 0.42–0.98). Other non-significant factors were being overweight at compilation time (OR: 1.44, 95% CI: 0.94–2.20, reference: being underweight or normal), female gender (OR: 1.39, 95% CI: 0.94–2.07), maintaining a vigorous physical activity (OR: 0.58, 95% CI: 0.33–1.02, reference: stopping the activity due to pandemic restrictions), maintaining hobbies/pastimes (OR: 0.57, 95% CI: 0.31–1.05, reference: never having hobbies/pastimes), and age from 50–59 years (OR: 0.59, 95% CI: 0.32–1.08, reference: age ≤ 39 years).

The multivariable logistic regression model showed that an increase in ESS score, increase in PHQ score, body weight modification, and, close to significance, having a dedicated room available for work were predictors of increased total PSQI score (Table 6). Worsening in the ESS score and the PHQ score were significantly interacted (*p* = 0.04).

## 3. Discussion

We here examined changes in sleep quantity and quality and daytime sleepiness in workers of an Italian public research organization who had transitioned from in-presence working to WH due to the COVID-19 pandemic. We found that the way WH impacted sleep was different depending on the baseline pre-WH sleep patterns, observing an improvement in sleep quality in poor sleepers and a worsening in good sleepers during WH.

Using the PSQI scale, we indeed did not find a meaningful difference in overall sleep quality (total PSQI score) between WH and pre-WH in the whole sample. However, similarly to other studies [39,44,45,54,55], we found that WH was associated with significantly longer sleep duration (+30 min). Furthermore, during WH, the percentage of participants with short sleep duration (≤6 h/night) significantly decreased, while that of participants reaching the recommended levels of sleep duration (>7 h/night) increased. To capture several sleep-related dimensions, in addition to sleep duration, we inspected other PSQI components and observed some significant changes during WH compared with pre-WH, i.e., sleep latency and sleep disturbances increased, which were in line with decreased sleep efficiency, while subjective sleep quality and daytime dysfunction improved. In our stratified analysis, lower sleep quality during WH was found among women in accordance with the literature on the adult population [56], while higher sleep quality before and during WH was reported among the researcher profile relative to technical and administrative staff, possibly due to the greater flexibility of researchers in the daily work schedule.

We found a significant impact of transition to WH on overall sleep quality only when we considered the baseline pre-WH levels of participants’ sleep quality. Indeed, subjects who were previously poor sleepers (total PSQI > 5) reported a significant increase in the quality of their sleep during WH. The improvement did not concern only the total PSQI score, but also its components: indeed, a significant percentage of poor sleepers reported an improvement in subjective sleep quality, sleep duration, and daytime dysfunction during WH. However, overall sleep quality decreased in the good sleepers, of whom a significant percentage experienced a worsening in subjective sleep quality, sleep latency, sleep efficiency, sleep disturbances, and use of sleep medications. Notably, as a clinically relevant perspective, the percentage of participants reporting an improvement in PSQI components was higher in the poor sleeper group than the percentage of participants reporting a deterioration of PSQI components in the good sleepers. Importantly, these results point out that the implications of WH for sleep are not the same for all subjects, and that the baseline sleep status of the subjects could be determinant for their sleep responses to WH, thus at least in part explaining the inconsistent literature data when they only considered the whole study population. It is possible that poor sleepers might have benefitted most from the transition to a new work arrangement and the associated changes in sleep–wake rhythms, as discussed below. Contrarily, good sleepers might have been adapted better to in-presence work so that the transition to WH negatively affected their sleep behavior and/or other potential factors associated with and impacting on sleep. Our findings are comparable to other studies concerning the impact of lockdowns on sleep and demonstrate that sleep quality could improve in some individuals and worsen in others during lockdown [57], and this could depend on the pre-pandemic sleep quality [58].

Prior to the COVID-19 pandemic, evidence on the impact of WH on sleep was limited and reported mixed results [39,47]. Most studies assessing the impact of WH on sleep were conducted in the context of the COVID-19 pandemic and in particular during the lockdown period(s). Evidence provided conflicting results regarding the effect of WH on overall sleep quality, as measured using the PSQI scale or other methods, with studies reporting no change [44,59] or a deterioration [46,48,54,60,61], potentially depending on the country-specific stringency of restrictive measures, the COVID-19 spread, as well as the assessment period(s) and study designs. Our subgroup analysis data, at least in poor sleepers, are consistent with other studies, including some conducted in Italy, showing an amelioration of sleep quality under WH during the COVID-19 pandemic [39,44,47,55,62]. Those who worked remotely during the pandemic not only slept more but also presented with a delayed bedtime and rise time and reduced social jet lag [44,54,55], [57]. Social jet lag is a risk factor for adverse health outcomes and impaired performance and derives from the mismatch between external (social and working) and internal (biological) sleep–wake timing [63]. Furthermore, the vulnerability of evening chronotype to poor sleep quality, shorter sleep duration, higher insomnia, and depression symptoms was moderated by and disappeared under remote working compared with in-office working [64]. If daily working time increased, WH was associated with better sleep quality and fewer insomnia symptoms compared with in-office working [55]. Under WH, the loosening of social and working obligations and the increased flexibility in the working schedule may favor a better reorganization of sleep–wake rhythms in accordance with individual’s circadian preference, thus potentially resulting in fewer sleep problems in some subjects.

However, many factors related to work (e.g., job position, workload, technology, work perceptions, organizational support), home (e.g., workstation, caring and domestic responsibility, distraction, commuting, connectivity), or individual and social background (e.g., individual preference/ability to WH, coping skills, mental and physical health, wellbeing, social support, lifestyle, demographics, work–family balance, work culture) can influence sleep during WH [65]. Several organizational factors of WH may affect health and wellbeing of workers: the sudden switching to WH as occurred during the pandemic led to a non-standardized workplace setting at home, where ergonomics, indoor environmental quality (lighting, noise, temperature, air quality), as well as operational, technical, and technological support could be suboptimal, thus ensuing low satisfaction, physical discomfort (e.g., low back pain, musculoskeletal disorder, etc.), stress and low concentration, and psychologically unsafe conditions, which may affect sleep [65,66,67]. Moreover, WH can trigger to blurred work–life boundaries compared with in-office work, resulting in higher working hours and workload (even at nights or early mornings), stress, anxiety, and family conflicts mostly in those living with cohabitants or children, and with low engagement with colleagues [65]. Other aspects that may affect sleep during WH could be type of housing (house versus apartment), or related to job position, prior experience with WH before the pandemic, interaction demands with colleagues/superiors as well as WH frequency [67].

In our study, the same factors that might contribute to the beneficial influence of WH on sleep in those having poor sleep might be perceived as disturbing in terms of sleep quality by those individuals, who were prior good sleepers and had difficulty in terms of sleep behavior in adapting to a new type of working. For instance, a flexible working schedule during WH could not be conducive to good sleep for all individuals. Blurred work–life boundaries may lead to higher workload and an increased screen based-technology use (e.g., computer, tablet, cellphone) during WH, also extending to bedtime, which may negatively affect sleep health and daytime functions [68]. In those who were prior good sleepers, the need to re-schedule work time may lead to different sleep–wake routines compared with their typical schedule dictated by their circadian preference as well as by in-office work and social conventions.

Interestingly, by comparing perceptions of pandemic-induced WH experience between two different countries (Korea and the Netherlands), Park et al. [67] found that a complex interplay of sociocultural backgrounds (e.g., collectivism/individualism, work culture) can influence the relationship between several potential predictive factors and physical/mental health (including sleep quality) and productivity during WH. Further studies are therefore warranted to disentangle the differential effects of WH on poor and good sleepers, and the potential causal and moderating factors.

Many of the behavioral and environmental factors potentially influencing sleep pattern during WH, as described above, are aspects of sleep hygiene, which include daily living practices that facilitate sleep and concern sleep environment (e.g., a dedicated and comfortable room for sleep), sleep schedule (e.g., sleep timing regularity), nightly routine, and daily habits (i.e., smoking, caffeine or alcohol intake and exercising before bedtime, in-bed use of electronic devices, in-bed working, eating dinner late, early afternoon napping, stress management) [69]. The utility of personalized sleep hygiene rules to improve sleep and promote health in nonclinical populations has been increasingly recognized [70], also during the COVID-19 pandemic [71], and demonstrated in different occupational sectors [25,72]. It would therefore be interesting to assess measures of sleep hygiene in workers who work remotely to understand the factors that affect their sleep behavior. In this regard, the study by Hrehova et al. [73] showed that sleep hygiene practice, as measured by the sleep hygiene index, was similar before and during WH in workers of different occupational sectors. However, in line with our findings of a differential impact of WH on sleep, WH was associated with malpractices in some aspects of sleep hygiene, but also improved sleep hygiene behavior in other aspects, depending on gender, household (living alone or with flat mates), or work-related factors, including workload, during WH [73].

EDS is considered an indicator of an person’s excessive average sleep propensity in daily life that impairs daytime physical and mental function, quality of life, and safety [74]. It can be caused by poor sleep quantity and quality, social jet lag [75], sleep disorders such as obstructive sleep apnea (OSA), as well as other chronic medical conditions and mental health disorders [74]. We found that the percentage of participants with EDS meaningfully decreased by 2.3% during WH compared with the in-presence working. This result is speculatively in keeping with better sleep quality and duration as we here observed in some subjects, and with reduced social jet lag as observed by others [44,54,55]. Interestingly, 13.2% of subjects increased napping behavior, which might contribute to counteracting and reducing EDS [76].

The multivariable logistic regression highlights factors associated with impairments of sleep quality during WH. In both poor and good sleepers, increased depression score was associated with worsened sleep quality. In line with our results, recent studies found that those experiencing a worsening in sleep during the pandemic situation also reported anxiety and deteriorations in mood [57,58]. Accordingly, there exists a close bidirectional association between depression and sleep disturbances so that one can be a risk factor for developing the other [14]. Abnormalities in sleep architecture including increased sleep latency, sleep fragmentation, and reduced restorative slow-wave sleep are frequently observed in patients with depression [77]. This might raise concern about workers’ health and wellbeing because most studies conducted during the COVID-19 pandemic reported that WH was associated with a negative impact on mental health, including anxiety and depression [5,49,66,78,79]. 

In good sleepers, not only increased depression severity, but also increased daytime sleepiness and body weight gain were associated with a worsening in sleep quality. EDS is a common symptom, which is linked to sleep disturbances not only in the clinical setting, but also in the general population because of insufficient sleep, circadian misalignment, or poor sleep hygiene (e.g., use of mobile phone before bedtime) [80]. We also found that depressive symptoms score significantly interacted with daytime sleepiness score. Basic pathophysiological mechanisms supporting this link include the induction of sleep fragmentation due to nighttime hyperarousal; dopaminergic, noradrenergic, and GABAergic hypoactivation; prefrontal cortex hypoconnectivity; longer circadian period; and light hyposensitivity [81]. These functional changes associated with depression can deteriorate sleep quality and lead to daytime sleepiness.

The relationship between body weight/adiposity and sleep disturbances, especially OSA, is well established, where both anatomical (recurrent narrowing and closure of the upper airway) as well as functional (increased inflammation, insulin resistance, diabetes, gastrointestinal disorders, poor diet quality) obesity-related conditions could disrupt normal sleep duration and/or quality [82]. As previously reported [49,83,84], WH can increase sedentary behavior and change lifestyle habits (i.e., diet, physical activity), which may lead to body weight gain, thus raising the risk for health-related outcomes including sleep impairment.

Regarding protective factors, in poor sleepers, morning chronotype was negatively associated with sleep impairment. This is in line with previous findings reporting a lower prevalence of sleep problems as well as mental issues in morning chronotype compared with the evening one [64,85,86]. In good sleepers, having a dedicated room for work was protective, albeit not statistically significantly, against a deterioration in sleep quality. This result is consistent with other findings suggesting that a suitable physical workspace, including a dedicated room for work and a good workstation setup, allows for more privacy, fewer distractions, better work performance, and improved work–family life balance [87], as well as better sleep quality and health in general [66,87].

This study has limitations, including the cross-sectional design, which prevents any causal inference; the choice of a convenience sample for the survey (i.e., employees of a research organization); and the lack of generalizability of the results to other employment sectors and/or types. Moreover, the results cannot represent the entire population sample, because only those subjects who experienced real changes during WH and hence remembered them would have been more prone to answer the survey. Those who did not remember relevant changes would have been less likely to participate in the survey. Another limitation is the possible recall bias due to retrospective self-reporting about pre-WH sleep pattern, which might be confounded by uncontrolled factors also considering the absence of the pandemic during the pre-WH period. Notably, in studies conducted during the early stages of the pandemic, it is impossible to separate the effects of COVID-19 lockdowns and associated uncertainties from the direct impacts of WH on sleep during this time. Our study was conducted almost 2 years after the introduction of WH in Italy (i.e., early March 2020). This might have led to a medium-term adaptation of workers to the WH arrangement beyond the pandemic context and far away from the restrictions, thus showing a seemingly direct effect of WH on sleep. Furthermore, the interpretation of the reason for the changes reported by participants can only be hypothetical, since it is impossible to separate the effects of the pandemic from those of the work reorganization. Among the first stressors, fear of the pandemic was potentially relevant, as well as a possible SARS-Cov-2 infection and the presence of persistent symptoms, so-called long COVID or post-COVID, which often included sleep disorders [33,34]. Unfortunately, the survey did not provide this information.

In our survey, we relied on validated questionnaires (PSQI, ESS) to assess self-reported sleep attributes, and used the global PSQI score to distinguish good and poor sleepers. Previous studies debated the ability of the global score of the PSQI (one-factor model) to accurately capture all sleep characteristics and proposed multifactorial models (two-factor or three-factor models) to improve PSQI performance in nonclinical populations [88,89]. Moreover, the structure validity of PSQI under the unique environmental, behavioral, and social factors associated with the COVID-19 pandemic and WH arrangements were not examined in our study. These contextual factors may affect sleep differently from “normal” situations [90]. Future studies could test the multifactorial structure of the PSQI in specific contexts like remote working. Of course, polysomnography is the gold standard for objective assessment of sleep and diagnosis of sleep disorders [20]. However, its limited accessibility and high resource and time demands make polysomnography impractical for population studies. More feasible, convenient, and accessible alternative recording devices should be used to provide an objective evaluation of sleep in research studies. One could be portable sleep monitoring such as respiratory polygraphy [91], which can be performed at home in a natural sleeping environment. This type of sleep recording device would be a useful tool for the evaluation of sleep under WH.

Despite its limitations, this study had the merit of shedding light on the different possible effects—improvement or worsening in sleep quality—that can be associated with a sudden change in the way of working. Moreover, studies like ours could have practical implications for both workplace policies and public health. Educational efforts should be undertaken to raise awareness among both employers and employees about the significance of sleep to health status, and to incorporate sleep evaluation and its related personal, environmental, and societal factors in health status assessment. This could be particularly important for remote working, where additional specific factors may emerge and be detrimental or advantageous on an individual basis. Knowledge of WH-associated risk or benefit on sleep would inform organizations and policy makers about strategies to provide and monitor adequate work environment and schedule/demands at home and promote sleep hygiene to improve workers’ overall health and wellbeing.

## 4. Materials and Methods

### 4.1. Study Design and Population

The present study belongs to a wider research project (“WorkInCovid”) evaluating the consequences of WH, which was largely introduced during the COVID-19 pandemic, on the health, wellbeing, work experience, and quality of life of full-time employees of the CNR [50]. A cross-sectional online survey was conducted among permanent workers at the CNR, which includes 4 professional profiles: with the exclusion of managers, workers are classified into: researcher (51%), technologist (9%), administrative (10%), and technician (27%). The first two profiles have greater organizational flexibility than the other two. Before the pandemic, CNR had not yet introduced WH and only a small percentage of workers (around 5%) were recurrent teleworkers.

### 4.2. Survey

Data were collected between 12 January 2022 and 9 March 2022. Using the LimeSurvey open source tool (Community Edition version 3.26.1), the survey collected information on sociodemographics (age, gender, educational level, professional profile, marital status, number of children, hometown and home dimensions, residence), work-related characteristics (number of days required to work at workplace during the WH period, home-to-work commuting time and means of transport before WH, type of workroom at home during WH, frequency of workstation sharing); and changes in lifestyle habits (physical activity during the WH period, sedentary behavior, hobbies or pastimes, diet). Information on chronotype—i.e., the subjective circadian orientation (evening, morning, and intermediate chronotype) about personal daily sleep–wake habits and the times of day of preference of certain activities—was collected using the reduced 5-item version of the Morningness–Eveningness Questionnaire (MEQr) [92,93]. Total scores of the 5-item rMEQ range from 4 to 26, whereby a higher score indicates a morning chronotype (classification: evening type: 4–10; intermediate: 11–18; morning type: 19–26). Depression symptoms were assessed through the validated Patient Health Questionnaire-9 (PHQ-9) [53], which is a 9-item index to assess the presence of depression symptoms. The questionnaire assesses how often the subjects had been disturbed by any of the 9 situations on a scale of 0 (not at all), 1 (for several days), 2 (at least half the time), and 3 (nearly every day). The total score ranks severity of depression, i.e., 0–4 (absent), 5–9 (mild), 10–14 (moderate), 15–19 (moderate-severe), and 20–27 (severe).

### 4.3. Questionnaires for Outcomes Assessment

Sleep quality and excessive daytime sleepiness (EDS) were evaluated through the Italian versions of validated questionnaires: the Pittsburgh Sleep Quality Index (PSQI) [51,52] and the Epworth Sleepiness Scale (ESS) [94,95], respectively. Two questions gathered information on napping habit (“Yes”, “No”), and the preferred napping time and duration. To determine whether the introduction of WH during the pandemic has changed workers’ sleep, PSQI and ESS questionnaires were administered twice, once retrospectively referring to the period before the introduction of WH and once referring to the time after the introduction of WH. We then measured differences in the outcome variables between periods before and during WH.

The PSQI is a widely used questionnaire evaluating seven different dimensions/components of sleep (subjective sleep quality, sleep latency, sleep duration, habitual sleep efficiency, sleep disturbances, use of sleep medications, and daytime dysfunction). Each dimension is scored between 0 (better) and 3 (worse). The global score is calculated by summing each component’s score, and can range from 0 to 21, where the higher the PSQI score, the worse the participants’ sleep quality. A score > 5 conventionally identifies poor sleepers. Among the PSQI components, sleep efficiency was calculated by dividing the reported number of hours slept by the reported number of hours spent in bed (determined from habitual bed and wake times) and multiplying by 100.

The ESS is an 8-item questionnaire to self-assess the general level of EDS. The questionnaire considers various situations from everyday life, and for each of them, the subject must establish the probability with which he/she might doze off or fall asleep. Each item is rated on a scale from 0 (“I have never dozed off”) to 3 (“high probability of dozing off”). The overall score is obtained by summing individual scores and can range from 0 to 24. Scores above 10 are indicative of increased daytime sleepiness compared to the physiological level (i.e., EDS), while scores above 15 are associated with abnormal daytime sleepiness.

### 4.4. Ethical Issues

This study was carried out according to the Declaration of Helsinki, positively evaluated by the CNR Data Protection Officer, and approved by the CNR Research Ethics and Integrity Committee on 28 October 2021 (Ethical Clearance number 0078918/2021). Participation was voluntary without compensation. All respondents provided online written consent by ticking a checkbox before answering the survey. Further details are reported in [49].

### 4.5. Statistical Analysis

Data were reported as counts and percentages for categorical variables, mean ± standard deviation, and median with interquartile interval for continuous variables. The two-tailed Wilcoxon signed-rank test was used to evaluate pre-WH versus during WH changes in the scores and the pseudomedian estimate of the location shift was computed. A non-parametric approach was used due to the ordinal nature of the scores. Checks for gaussian distribution (Shapiro–Wilks test) were rejected often, due to the presence of outliers and asymmetry in the score distributions. When the deviation from normality was significant, we indicated the median value and preferentially used non-parametric tests, even if the sample size allowed for us to use also traditional parametric tests, according to Lumley et al. [96]. The McNemar test and the Stuart–Maxwell test were used to compare discrete distributions before and during the WH period for categorical variables. Sleep and daytime sleepiness worsening/improvement were defined as an increase/decrease in at least one point in the transition to the WH period of PSQI, PSQI components, and ESS, respectively.

Uni- and multivariable logistic regressions were performed to identify the potential risk factors for sleep worsening. A backward algorithm based on the Akaike Information Criterion and clinical considerations were used for model selection. The presence of interaction terms was tested by the Likelihood Ratio test [97]. We adjusted for confounders and calculated the odds ratios (ORs), the 95% confidence intervals (CIs), and the corresponding *p*-values. Values of p < 0.05 were considered statistically significant. All analyses were performed using the statistical software program R (version 4.2.0) [98].

## 5. Conclusions

Conclusively, our study provided evidence of a non-uniform impact of sudden and forced introduction of WH on sleep, depending on individual variability and especially the baseline sleep quality of workers on top of the pandemic background. During WH, poor sleepers responded in a positive way, improving their sleep quality, while good sleepers reported a worsening in sleep quality. These results need to be confirmed and expanded by further longitudinal studies. However, it could be important to raise hypotheses on the influence of home working with respect to in-office working on health: this knowledge would ultimately contribute to defining remote working policy that takes into consideration the individual variability in responses to life and work changes and minimizes health risks through targeted sleep health promotion. This could be most important if WH is to continue increasingly in the future, and the benefits of WH for sleep and overall public health for some workers should be maximized, while addressing its potential negative consequences for others.

## Figures and Tables

**Figure 1 clockssleep-07-00013-f001:**
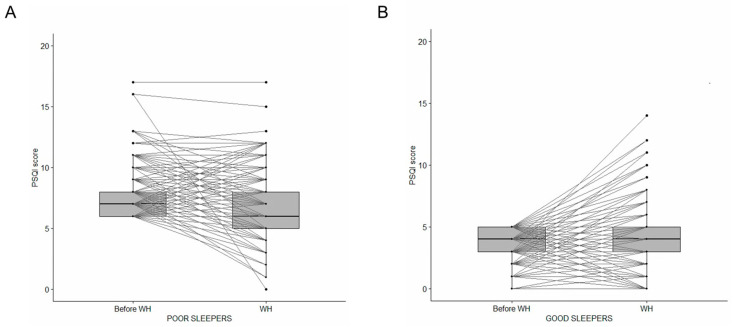
PSQI total score before and during WH in participants reporting poor sleep (**A**) and good sleep (**B**) before the transition to WH. An improvement in sleep quality was reported during WH by poor sleepers (**A**), while a worsening in sleep quality was reported by good sleepers (**B**).

**Table 1 clockssleep-07-00013-t001:** Participants’ characteristics.

	Sample (*n* = 748)	
	*n*	%
Gender		
Men	317	42.4
Women	431	57.6
Age (years)		
≤39	90	12.0
40–49	275	36.8
50–59	285	38.1
≥60	98	13.1
Cohabitation		
Living alone	108	14.4
Not living alone, no children	282	37.7
Not living alone, with children	358	47.9
Place of residence ^a^		
North	244	32.6
Center	261	34.9
South	168	22.5
Islands	75	10.0
Education		
Graduation	614	82.1
No graduation	134	17.9
Professional profile		
Administrative and technical staff	238	31.8
Researcher and technologist	510	68.2

^a^ North: Aosta Valley, Emilia-Romagna, Friuli-Venezia Giulia, Liguria, Lombardy, Piedmont, Trentino-Alto Adige, and Veneto. Center: Lazio, Marche, Tuscany, and Umbria. South: Abruzzo, Apulia, Basilicata, Calabria, Campania, and Molise. Islands: Sardinia and Sicily.

**Table 2 clockssleep-07-00013-t002:** Sleep parameters before WH (pre-WH) and during WH. Summaries of location, scale, and percentages of PSQI and ESS, sleep duration, and sleep efficiency at the two different times and differences between during-WH and pre-WH are shown. The *p*-values of test on either difference of scores or changes in the distributions of categorical variables are presented.

	Pre-WH	During WH	Difference	*p*-Value ^1^
PSQI score				
Mean ± SD	5.25 ± 2.39	5.15 ± 2.57	−0.10 ± 2.19	n.a. ^2^
Median, IQR	5, 4–6	5, 3–7	0, (−1)–1	0.04
Poor sleepers (PSQI > 5) (n [%])	285 [38.7]	272 [36.9]	--	0.37
Good sleepers (PSQI ≤ 5) (n [%])	452 [61.3]	465 [63.1]	--	0.37
Sleep duration				
Min (mean ± SD)	403 ± 56	416 ± 59	13.37 ± 51.7	n.a.
Min (median, IQR)	420, 360–420	420, 360–480	0, 0–60	<0.001
≤6 h (n [%])	249 [33.8]	201 [27.3]	--	<0.001
7–8 h (n [%])	484 [65.7]	519 [70.4]	--	
≥9 h (n [%])	4 [0.5]	17 [2.3]	--	
Sleep efficiency				
Mean% ± SD	89.2 ± 10.1	88.0 ± 10.9	−1.2 ± 9.23	n.a.
Median%, IQR	91.7, 83.5–100.0	88.9, 82.3–96.6	0, (−5.2)–3.3	0.004
ESS score				
Mean ± SD	4.46 ± 3.20	4.30 ± 2.85	−0.16 ± 2.03	n.a.
Median, IQR	4, 2–6	4, 2–6	0, 0–1	0.16
EDS (ESS > 10) (n [%])	40 [5.4]	23 [3.1]	--	0.007

^1^ Paired Wilcoxon test for scores, either McNemar or Stuart–Maxwell test for 2 × 2 contingency tables. ^2^ n.a.: not applicable (*t*-test on the mean difference was not applied due to the non-normal distribution). PSQI: Pittsburgh Sleep Quality Index; SD: standard deviation; IQR: interquartile range; ESS: Epworth Sleepiness Scale; EDS: excessive daytime sleepiness.

**Table 3 clockssleep-07-00013-t003:** Percentages of subjects in the whole sample with stable, improved, and worsened PSQI components during WH.

PSQI Components	Stable (%)	Improved (%)	Worsened (%)
Sleep quality	72.0	16.6	11.4
Sleep latency	75.6	8.4	16.0
Sleep duration	59.7	27.7	12.6
Sleep efficiency	70.7	13.4	15.9
Sleep disturbances	83.4	5.3	11.3
Use of sleep medication	91.0	3.7	5.3
Daytime dysfunction	75.3	14.4	10.3

**Table 4 clockssleep-07-00013-t004:** Association between the proportions of subjects among poor sleepers and good sleepers with meaningful changes in PSQI components during WH.

PSQI Component	No Meaningful Change	Meaningful Change ^1^		*p*-Value
	(n)	(n)	(%)	
Sleep quality				
Poor sleepers	228	57	20.0	<0.001
Good sleepers	415	37	8.2	
Sleep latency				
Poor sleepers	267	18	6.3	0.59
Good sleepers	429	23	5.1	
Sleep duration				
Poor sleepers	216	69	24.2	<0.001
Good sleepers	410	42	9.3	
Sleep efficiency				
Poor sleepers	263	19	7.7	0.10
Good sleepers	435	17	3.8	
Sleep disturbances				
Poor sleepers	260	25	8.8	0.55
Good sleepers	405	47	10.4	
Use of sleep medication				
Poor sleepers	274	11	3.9	0.38
Good sleepers	441	11	2.4	
Daytime dysfunction				
Poor sleepers	236	49	17.2	<0.001
Good sleepers	416	36	8.0	

^1^ In the subgroup of poor sleepers, improved condition (higher component score); in the subgroup of good sleepers, worsened condition (lower component score).

**Table 5 clockssleep-07-00013-t005:** Multivariable logistic regression model defining the independent predictors of increased PSQI score within 285 poor sleepers. An increase in depressive symptoms and the morning chronotype were risk and protective factors for sleep worsening, respectively.

Variable	OR	LB—95% CI	UB—95%CI
Worsening in PHQ-9 score	3.64 *	2.04	6.60
Circadian type			
Evening type	1.00	--	--
Intermediate type	0.62	0.26	1.51
Morning type	0.34 *	0.12	0.95

PHQ-9: Patient Health Questionnaire-9; OR: odds ratio; LB: lower bound; UB: upper bound. * *p* < 0.05.

**Table 6 clockssleep-07-00013-t006:** Multivariable logistic regression model defining the independent predictors of increased PSQI score within 452 good sleepers. An increase in depressive symptoms, daytime sleepiness score, and body weight were significant predictors of sleep worsening.

Variable	OR	LB—95% CI	UB—95%CI
Worsening in PHQ-9 score	4.26 *	2.50	7.32
Worsening in ESS score	2.47 *	1.35	4.49
Weight variation			
Unchanged weight	1.00	--	--
Decreased weight	1.88	0.97	3.58
Increased weight	2.55 *	1.62	4.06
Having a dedicated room available for work	0.66	0.41	1.05
Worsening in ESS score ^1^	0.961	0.49	1.86

^1^ Adjusting for worsening in PHQ-9 score. PHQ-9: Patient Health Questionnaire-9; ESS: Epworth Sleepiness Scale. OR: odds ratio; LB: lower bound; UB: upper bound. * *p* < 0.05.

## Data Availability

Data available on request due to privacy/ethical restrictions.

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
