# Peer review of "Not All Workers Experience Equal Sleep Changes: Insights from the “WorkInCovid” Project"

_2624-5175, 2025, doi:10.3390/clockssleep7010013_

Round 1
Reviewer 1 Report
Comments and Suggestions for Authors
This is an interesting manuscript on the impact of working from home on sleep in the context of COVID-19. Findings from this survey of workers demonstrate the impact of working from home on sleep, both positive impacts and negative impacts. It is clear from this paper that COVID-19 and the changes to work has impacted sleep health. Some feedback to consider is listed below for each section:
Intro:
Overall the introduction is well-written and presents the story of this paper well. I believe some discussion of population sleep health prior to COVID-19 would strengthen this discussion, for example working adults were typically not receiving enough sleep (under 7 hours) and poor quality sleep prior to COVID-19 and some of the factors that caused this. This is important to know, especially when hypothesising that working from home may actually improve sleep.
Methods:
Methods section is clear and well described. With the statistical analyses, why was the linear regression done to predict sleep worsening, when the introduction stated that clear hypotheses on whether sleep would worsen or improve could not be understood from the literature. There appears to be a disconnect between the introduction and sleep possibly improving, but then the statistical analysis being to determine why sleep worsened.
Results:
Well presented and clear. Interesting findings!
Discussion:
Was it possible that participants who’s sleep worsened had COVID-19 or long COVID? This potentially impacted sleep findings over and above that of the work set-up.
Good discussion overall. I suggest including some discussion of sleep hygiene principles as many of the factors discussed that could improve or worsen sleep (e.g. working late into the evening, having a separate room for sleep, daytime napping) are aspects of sleep hygiene. Future research could include measures of sleep hygiene in workers that work remotely to understand the factors that influence sleep.
Comments on the Quality of English Language
The quality is good overall but some instances of word choice and sentence structure could be improved.
Author Response
Reviewer #1
This is an interesting manuscript on the impact of working from home on sleep in the context of COVID-19. Findings from this survey of workers demonstrate the impact of working from home on sleep, both positive impacts and negative impacts. It is clear from this paper that COVID-19 and the changes to work has impacted sleep health. Some feedback to consider is listed below for each section:
Intro:
Overall the introduction is well-written and presents the story of this paper well. I believe some discussion of population sleep health prior to COVID-19 would strengthen this discussion, for example working adults were typically not receiving enough sleep (under 7 hours) and poor quality sleep prior to COVID-19 and some of the factors that caused this. This is important to know, especially when hypothesising that working from home may actually improve sleep.
Response: We appreciate the reviewer's suggestion. Accordingly, in the Introduction we have elaborated on the increasing burden of sleep deprivation and poor sleep quality in modern societies, and especially in working populations, before the COVID-19 pandemic, along with the potential causal factors.
Methods:
Methods section is clear and well described. With the statistical analyses, why was the linear regression done to predict sleep worsening, when the introduction stated that clear hypotheses on whether sleep would worsen or improve could not be understood from the literature. There appears to be a disconnect between the introduction and sleep possibly improving, but then the statistical analysis being to determine why sleep worsened.
Response: The literature findings about changes in sleep pattern are heterogeneous and often inconsistent, because of several factors including differences in study design, periods, and populations. For the regression analysis, we had to formulate a hypothesis on either improvement or worsening of sleep, and our (research and medical) interest on factors that might drive sleep impairment and were worthier of attention and interventions by employers, employees and public health authorities led us to use sleep worsening as the independent variable.
Results:
Well presented and clear. Interesting findings!
Discussion:
Was it possible that participants who’s sleep worsened had COVID-19 or long COVID? This potentially impacted sleep findings over and above that of the work set-up.
Response: Unfortunately, the survey did not provide this information. We have included this point in the limitations of the study.
Good discussion overall. I suggest including some discussion of sleep hygiene principles as many of the factors discussed that could improve or worsen sleep (e.g. working late into the evening, having a separate room for sleep, daytime napping) are aspects of sleep hygiene. Future research could include measures of sleep hygiene in workers that work remotely to understand the factors that influence sleep.
Response: We thank the reviewer for the comment, which gave us the opportunity to improve the discussion by elaborating on sleep hygiene practices and their relevance to remote working.
Comments on the Quality of English Language
The quality is good overall but some instances of word choice and sentence structure could be improved.
Response: The manuscript has been edited to improve English.
Reviewer 2 Report
Comments and Suggestions for Authors
I read with interest the manuscript “Working from home is not equal for everyone’s sleep: the 2 'WorkInCovid' project”. In my opinion it addresses a highly relevant topic—how working from home during the COVID-19 pandemic influenced sleep health. The authors used validated tools, such as the Pittsburgh Sleep Quality Index (PSQI) and the Epworth Sleepiness Scale (ESS). It is worth mentioning that nowadays the study is well-timed, taken into consideration the global shift towards remote work, and the paper contributes valuable data to the literature on sleep health and occupational well-being. However I have some suggestions for improvement:
The title is engaging but could be refined to provide clearer information about the study's focus. For example, "Not all workers experience equal sleep changes: Insights from the 'WorkInCovid' project" better conveys that the research specifically examines differences in how working from home impacts sleep quality and patterns. The current title, "Working from home is not equal for everyone’s sleep," is somewhat ambiguous, as "is not equal" does not explicitly indicate whether the focus is on sleep quality, duration, or other aspects of sleep.
The cross-sectional design is appropriately justified, but the retrospective nature of pre-WH sleep assessments introduces recall bias. I suggest that the authors emphasize this limitation more explicitly and discuss its potential impact on the results.
The discussion effectively interprets the findings in light of existing literature. However I would suggest to expand on the practical implications of the findings, especially for workplace policies and public health interventions.
Tables and figures are informative but could benefit from brief descriptive captions summarizing key findings.
The study's reliance on validated questionnaires such as PSQI and ESS is commendable and ensures robust data collection. However, it is worth noting that full polysomnography (PSG) remains the gold standard for the comprehensive assessment of sleep quality. PSG provides detailed insights into sleep architecture, including critical parameters like sleep fragmentation, which has been linked to cardiovascular risk (DOI: 10.17219/dmp/185395). While PSG is highly informative, it is also expensive, resource-intensive, and impractical for large-scale studies such as this. In this context, portable respiratory polygraphy offers a promising alternative due to its accessibility and feasibility for use even in home settings. I would suggest to cover also this topic in discussion section.
Furthermore, the introduction mentions the broad health consequences of sleep disturbances. It is important to emphasize that conditions like obstructive sleep apnea and sleep bruxism—the latter being the second most prevalent sleep-related movement disorder—are associated with heightened systemic inflammation and oxidative stress. These mechanisms significantly increase cardiovascular risk, underscoring the relevance of addressing sleep health. Recent findings support this association, highlighting the link between sleep bruxism, chronic inflammation, and altered sleep architecture (DOI: 10.1038/s41598-025-86833-y).
I congratulate the authors on an interesting and valuable study. I trust that the suggested revisions will contribute to enhancing the quality of the manuscript. This work represents a significant contribution to the field of sleep medicine, and I look forward to reviewing the revised version with great interest.
Author Response
Reviewer #2
I read with interest the manuscript “Working from home is not equal for everyone’s sleep: the 2 'WorkInCovid' project”. In my opinion it addresses a highly relevant topic—how working from home during the COVID-19 pandemic influenced sleep health. The authors used validated tools, such as the Pittsburgh Sleep Quality Index (PSQI) and the Epworth Sleepiness Scale (ESS). It is worth mentioning that nowadays the study is well-timed, taken into consideration the global shift towards remote work, and the paper contributes valuable data to the literature on sleep health and occupational well-being. However I have some suggestions for improvement:
The title is engaging but could be refined to provide clearer information about the study's focus. For example, "Not all workers experience equal sleep changes: Insights from the 'WorkInCovid' project" better conveys that the research specifically examines differences in how working from home impacts sleep quality and patterns. The current title, "Working from home is not equal for everyone’s sleep," is somewhat ambiguous, as "is not equal" does not explicitly indicate whether the focus is on sleep quality, duration, or other aspects of sleep.
Response: We greatly appreciate the reviewer’s suggestion. The title has been changed accordingly.
The cross-sectional design is appropriately justified, but the retrospective nature of pre-WH sleep assessments introduces recall bias. I suggest that the authors emphasize this limitation more explicitly and discuss its potential impact on the results.
Response: As the reviewer highlighted, in the manuscript we acknowledged the limitation of the recall bias. Accordingly, we have expanded the issue in the Discussion.
The discussion effectively interprets the findings in light of existing literature. However I would suggest to expand on the practical implications of the findings, especially for workplace policies and public health interventions.
Response: We fully agree with the reviewer. Practical implications of our findings have been now expanded in the Discussion session.
Tables and figures are informative but could benefit from brief descriptive captions summarizing key findings.
Response: Figures and Tables captions have been modified accordingly.
The study's reliance on validated questionnaires such as PSQI and ESS is commendable and ensures robust data collection. However, it is worth noting that full polysomnography (PSG) remains the gold standard for the comprehensive assessment of sleep quality. PSG provides detailed insights into sleep architecture, including critical parameters like sleep fragmentation, which has been linked to cardiovascular risk (DOI: 10.17219/dmp/185395). While PSG is highly informative, it is also expensive, resource-intensive, and impractical for large-scale studies such as this. In this context, portable respiratory polygraphy offers a promising alternative due to its accessibility and feasibility for use even in home settings. I would suggest to cover also this topic in discussion section.
Response: We thank the reviewer for the comment. The suggested topic has now been discussed.
Furthermore, the introduction mentions the broad health consequences of sleep disturbances. It is important to emphasize that conditions like obstructive sleep apnea and sleep bruxism—the latter being the second most prevalent sleep-related movement disorder—are associated with heightened systemic inflammation and oxidative stress. These mechanisms significantly increase cardiovascular risk, underscoring the relevance of addressing sleep health. Recent findings support this association, highlighting the link between sleep bruxism, chronic inflammation, and altered sleep architecture (DOI: 10.1038/s41598-025-86833-y).
Response: In agreement with the useful reviewer’s comments, we have elaborated on this topic in the Introduction.
I congratulate the authors on an interesting and valuable study. I trust that the suggested revisions will contribute to enhancing the quality of the manuscript. This work represents a significant contribution to the field of sleep medicine, and I look forward to reviewing the revised version with great interest.
Reviewer 3 Report
Comments and Suggestions for Authors
Thank you for the opportunity to review the manuscript “Working from home is not equal for everyone’s sleep: the ‘WorkInCovid’ project”, which was aimed to examine “the association of WH during the pandemic with sleep health in workers of a public research organization”.
The topic is interesting. However, several revisions are required to consider publication.
MAJOR COMMENTS
- The objective and the design of the study are not clearly identified.
- The use of statistical descriptors and tests according to data distribution is not consistent (either parametric or non-parametric), implying the risk of bias and inappropriate inference. Statistician advice is recommended.
- According to the 95% C.I. described in the results section, the interpretation of some of the results is controversial.
- The individual history of illness due to COVID-19 that may modify the outcome variable was not included in the study.
- The individual history of comorbidities that could influence the outcome variable was not included in the description and the analysis.
- The independent use and interpretation of the constructs included in the sleep quality index has not been validated. However, the factor structure of the instrument on non-clinical populations supports the use of three factors (i.e., sleep efficiency, sleep latency, and sleep quality) (Jia et al. Sleep and Biological Rhythms 2019; 17:209–221.
- The effects of the interaction among variables is not described.
MINOR COMMENTS
- The Introduction is mainly focused on the reasons to work from home, instead of on sleep habits and the variables that could influence quality of sleep.
- The first two paragraphs of the Discussion section could be included in the Introduction section, while a summary of results could be provided at the beginning of the Discussion section.
- The text requires edition (e.g. very long sentences) and revision of typing errors (e.g. double symbols and inappropriate hyphening), as well as missing data (Table S3).
The text requires edition (e.g. very long sentences) and revision of typing errors (e.g. double symbols and inappropriate hyphening), as well as missing data (Table S3).
Author Response
Reviewer #3
Thank you for the opportunity to review the manuscript “Working from home is not equal for everyone’s sleep: the ‘WorkInCovid’ project”, which was aimed to examine “the association of WH during the pandemic with sleep health in workers of a public research organization”.
The topic is interesting. However, several revisions are required to consider publication.
MAJOR COMMENTS
- The objective and the design of the study are not clearly identified.
Response: We thank the reviewer who gave us the opportunity to explain in more depth the aim of our work, which we had already indicated in the previous manuscript at lines 127-148. We then added the following text:
“The aim of this work was to verify what effect teleworking had on sleep during the COVID-19 lockdown period. The hypotheses we derived from our observations on workers were that:
- Workers who had a better quality of sleep before the lockdown could have a worsening of the quality of sleep, due to the combined effect of anxiety related to the pandemic and the sudden and unorganized change in working methods;
- Workers who had poor quality sleep before the lockdown could have an improvement in sleep during teleworking, due to the reorganization of time dedicated to work and family and the avoidance of commuting.
We also assumed that workers who had experienced a change in their sleep habits would be the most willing to respond, while those who had not noticed any changes in their sleep would be less interested in our survey.”
- The use of statistical descriptors and tests according to data distribution is not consistent (either parametric or non-parametric), implying the risk of bias and inappropriate inference. Statistician advice is recommended.
Response: The distribution of the scores obtained from the questionnaires was initially studied using mean, median, and standard deviation. The Kolmogorov-Smirnov and Shapiro-Wilk tests were used to verify whether the distribution was normal. When the deviation from normality was significant, we indicated the median value and we preferentially used non-parametric tests, even if the sample size allowed us to use also traditional parametric tests, according to Lumley et al. (2002). The statistical methodology is detailed in Section 2.5; this section, and the results of the study, were edited by three of the authors (A.B., G.L.T., and N.M.) who are statisticians or teach medical statistics and research statistics.
Lumley, T.; Diehr, P.; Emerson, S.; Chen, L. The importance of the normality assumption in large public health data sets. Annu.Rev. Public Health 2002, 23, 151–169.
- According to the 95% C.I. described in the results section, the interpretation of some of the results is controversial.
Response: We appreciate the reviewer's clarification. Several variables showed non-significant variations. The explanation we had adopted in the first draft of the manuscript could have misled readers; we therefore preferred to make explicit which variations were significant and which, vice versa, did not reach significance.
- The individual history of illness due to COVID-19 that may modify the outcome variable was not included in the study.
Response: Unfortunately, the survey did not provide this information. We have included this point in the limitations of the study.
- The individual history of comorbidities that could influence the outcome variable was not included in the description and the analysis.
Response: As previously published in another paper on the same study population (Scoditti et al, PLoS One. 2024), comorbidities were present in 12.9% of participants and did not influence the outcomes variables assessed in the present manuscript, hence we did not include it in the analysis.
Scoditti E, Bodini A, Sabina S, Leo CG, Mincarone P, Rissotto A, Fusco S, Guarino R, Ponzini G, Tumolo MR, Magnavita N, Tripepi GL, Garbarino S. Effects of working from home on lifestyle behaviors and mental health during the COVID-19 pandemic: A survey study. PLoS One. 2024 Apr 1;19(4):e0300812. doi: 10.1371/journal.pone.0300812
- The independent use and interpretation of the constructs included in the sleep quality index has not been validated. However, the factor structure of the instrument on non-clinical populations supports the use of three factors (i.e., sleep efficiency, sleep latency, and sleep quality) (Jia et al. Sleep and Biological Rhythms 2019; 17:209–221).
Response: We greatly appreciate the reviewer's suggestion and believe that taking into account the multifactorial structure of the PSQI could add elements of interest. We considered this as a possible development of the research. We added this suggestion to the manuscript.
- The effects of the interaction among variables is not described.
Response: We thank very much the reviewer for the important note. A discussion on the interaction between ESS and PHQ scores found among good sleepers has been added in the Discussion.
MINOR COMMENTS
The Introduction is mainly focused on the reasons to work from home, instead of on sleep habits and the variables that could influence quality of sleep.
Response: In agreement with the reviewer’s comment, the Introduction has been expanded to include further elaboration on sleep.
The first two paragraphs of the Discussion section could be included in the Introduction section, while a summary of results could be provided at the beginning of the Discussion section.
Response: We thank the reviewer for the suggestion. The first two paragraphs of the Discussion have been deleted (arguments already covered in the Introduction), and a brief summary of main results has been added at the beginning.
The text requires edition (e.g. very long sentences) and revision of typing errors (e.g. double symbols and inappropriate hyphening), as well as missing data (Table S3).
Response: The manuscript has been edited to be more fluent and correct. We apologize for the missing Table S3. It has been now reuploaded.
Reviewer 4 Report
Comments and Suggestions for Authors
Your manuscript is a timely and relevant one: it addresses the heterogeneous effects of working from home (WH) on sleep quality, particularly distinguishing between poor and good sleepers. The paper is well-organized overall, the methodology is sound, and your primary finding—that baseline sleep status strongly moderates the effects of WH—is a valuable contribution. The study aligns well with Clocks & Sleep’s focus on circadian rhythms, sleep, and health.
The manuscript offers a unique perspective by emphasizing baseline sleep patterns, highlighting that pre-teleworking sleep quality is crucial for understanding the diverse ways individuals adapt to working from home. This nuanced approach helps clarify the conflicting evidence in the literature regarding remote work’s impact on sleep. Additionally, the study’s methodological rigor is evident through the use of validated instruments such as the PSQI and ESS, a careful classification of participants into “poor sleepers” and “good sleepers,” and the application of backward logistic regression to accurately identify key predictors. The findings also hold significant public health relevance, underscoring the importance of targeted sleep-promotion strategies—particularly for workers who were previously good sleepers but experienced sleep deterioration when transitioning to remote work.
Points for Improvement
Contextualizing Findings in Recent Work-from-Home Literature:
While your references are generally up-to-date, incorporating newly published studies could further enhance your discussion on why certain subgroups respond differently to WH. For example, Cheung (2024) explores practical organizational strategies (e.g., ergonomics, boundary-setting) that can mitigate negative sleep outcomes for individuals accustomed to working on-site. Similarly, Hrehová et al. (2024) found that while total Sleep Hygiene Index scores remain stable, specific sleep hygiene elements tend to worsen during WH. Briefly integrating such recent findings into the Discussion section would reinforce your argument about the differential impact on poor versus good sleepers.
Further Detail on “Good Sleepers” Deterioration:
You note that participants who were good sleepers before the pandemic experienced a measurable decline in PSQI components after WH began. While you provide plausible psychosocial and environmental explanations (e.g., weight gain, depression, lack of a dedicated workspace), expanding on potential physiological or circadian mismatches—such as increased nighttime screen exposure or blurred work-life boundaries—could strengthen the mechanistic clarity of your findings.
Deepening the Discussion of Socio-Contextual Factors:
While your discussion touches on mental health (e.g., depression), body weight changes, and circadian typology, you could further explore whether gender norms, caregiving responsibilities, or job demand variations mediate or moderate these effects. For instance, some studies (e.g., Park et al., 2024) suggest that national and regional contexts influence how workers adapt to WH, including changes in daily routines, childcare responsibilities, and commuting time. Adding a brief paragraph on these socio-contextual factors would enhance the breadth of your discussion.
Additional References:
Cheung, V.K.L. (2024). Practical Considerations of Workplace Wellbeing Management in Post-Pandemic Work-from-Home Scenarios. International Journal of Environmental Research and Public Health, 21(7), 924.
Hrehová, L., Bušková, J., Seifert, B., Mezian, K. (2024). Working from Home is Linked to Altered Sleep Hygiene Behavior. Work, 77(4), 1135–1142.
Park, S.Y., Lee, R., Newton, C., Han, G. (2024). How are People Coping with Working from Home during the COVID-19 Pandemic? PLoS ONE, 19(4), e0301351.
Author Response
Reviewer #4
Your manuscript is a timely and relevant one: it addresses the heterogeneous effects of working from home (WH) on sleep quality, particularly distinguishing between poor and good sleepers. The paper is well-organized overall, the methodology is sound, and your primary finding—that baseline sleep status strongly moderates the effects of WH—is a valuable contribution. The study aligns well with Clocks & Sleep’s focus on circadian rhythms, sleep, and health.
The manuscript offers a unique perspective by emphasizing baseline sleep patterns, highlighting that pre-teleworking sleep quality is crucial for understanding the diverse ways individuals adapt to working from home. This nuanced approach helps clarify the conflicting evidence in the literature regarding remote work’s impact on sleep. Additionally, the study’s methodological rigor is evident through the use of validated instruments such as the PSQI and ESS, a careful classification of participants into “poor sleepers” and “good sleepers,” and the application of backward logistic regression to accurately identify key predictors. The findings also hold significant public health relevance, underscoring the importance of targeted sleep-promotion strategies—particularly for workers who were previously good sleepers but experienced sleep deterioration when transitioning to remote work.
Points for Improvement
Contextualizing Findings in Recent Work-from-Home Literature:
While your references are generally up-to-date, incorporating newly published studies could further enhance your discussion on why certain subgroups respond differently to WH. For example, Cheung (2024) explores practical organizational strategies (e.g., ergonomics, boundary-setting) that can mitigate negative sleep outcomes for individuals accustomed to working on-site. Similarly, Hrehová et al. (2024) found that while total Sleep Hygiene Index scores remain stable, specific sleep hygiene elements tend to worsen during WH. Briefly integrating such recent findings into the Discussion section would reinforce your argument about the differential impact on poor versus good sleepers.
Response: We appreciate the reviewer’s suggestions, which allowed us to improve the Discussion with updated literature.
Further Detail on “Good Sleepers” Deterioration:
You note that participants who were good sleepers before the pandemic experienced a measurable decline in PSQI components after WH began. While you provide plausible psychosocial and environmental explanations (e.g., weight gain, depression, lack of a dedicated workspace), expanding on potential physiological or circadian mismatches—such as increased nighttime screen exposure or blurred work-life boundaries—could strengthen the mechanistic clarity of your findings.
Response: In agreement with the reviewer’s comment, we have expanded the discussion regarding potential factors, including physiological or circadian mismatches, that might contribute to sleep impairment in good sleepers.
Deepening the Discussion of Socio-Contextual Factors:
While your discussion touches on mental health (e.g., depression), body weight changes, and circadian typology, you could further explore whether gender norms, caregiving responsibilities, or job demand variations mediate or moderate these effects. For instance, some studies (e.g., Park et al., 2024) suggest that national and regional contexts influence how workers adapt to WH, including changes in daily routines, childcare responsibilities, and commuting time. Adding a brief paragraph on these socio-contextual factors would enhance the breadth of your discussion.
Additional References:
Cheung VKL. Practical Considerations of Workplace Wellbeing Management under Post-Pandemic Work-from-Home Conditions. Int J Environ Res Public Health. 2024 Jul 15;21(7):924.
Hrehová, L., Bušková, J., Seifert, B., Mezian, K. (2024). Working from Home is Linked to Altered Sleep Hygiene Behavior. Work, 77(4), 1135–1142.
Park, S.Y., Lee, R., Newton, C., Han, G. (2024). How are People Coping with Working from Home during the COVID-19 Pandemic? PLoS ONE, 19(4), e0301351.
among those to be addressed.
Response: The reviewer’s issue has allowed us to discuss some important points, which could strengthen the interpretation of our findings. We appreciate very much the suggestions.
Round 2
Reviewer 3 Report
Comments and Suggestions for Authors
The manuscript was substantially improved, now it can stand by its own rigth.
Just revision for minor editing is required, for example:
- Line 176, missing word "of", or line 653 to correct "no uniform" for "non unifom");
- To imporve punctuation in the Conclusion Section, to assisst the reader to follow the ideas.
- Table S3 requieres a symbol in the empty boxes, to guide the reader to the clarification for not showing the data.
Comments on the Quality of English LanguageRevision for minor editing is required, for example:
- Line 176, missing word "of", or line 653 to correct "no uniform" for "non unifom");
- To imporve punctuation in the Conclusion Section, to assisst the reader to follow the ideas.
- Table S3 requieres a symbol in the empty boxes, to guide the reader to the clarification for not showing the data.
Author Response
Comment #2
The manuscript was substantially improved, now it can stand by its own rigth.
Just revision for minor editing is required, for example:
- Line 176, missing word "of", or line 653 to correct "no uniform" for "non unifom");
Response: We thank the reviewer for the note. The corrections have been made.
- To imporve punctuation in the Conclusion Section, to assisst the reader to follow the ideas.
Response: We thank the reviewer for the note. The punctuation has been modified to improve readability of the Conclusions.
- Table S3 requieres a symbol in the empty boxes, to guide the reader to the clarification for not showing the data.
Response: We agree with the reviewer: for better clarity, in Table S3 we have reported “d.n.r.” (data not reported) and the symbol for the related explanation.